# A Portable Laser Spectroscopic System for Measuring Nitrous Oxide Emissions on Fertilized Cropland

**DOI:** 10.3390/s23156686

**Published:** 2023-07-26

**Authors:** Gerrit Stiefvater, Yvonne Hespos, Dominic Wiedenmann, Armin Lambrecht, Raimund Brunner, Jürgen Wöllenstein

**Affiliations:** 1Fraunhofer Institute for Physical Measurement Techniques IPM, Georges-Köhler-Allee 301, 79110 Freiburg, Germanyjuergen.woellenstein@ipm.fraunhofer.de (J.W.); 2Laboratory for Gas Sensors, Department of Microsystems Engineering-IMTEK, University of Freiburg, Georges-Köhler-Allee 102, 79110 Freiburg, Germany

**Keywords:** cropland, fertilizer, gas emission, greenhouse gases, laser spectroscopy, mid infrared, N_2_O, nitrous oxide, portable system

## Abstract

Nitrous oxide (laughing gas, N_2_O) is a relevant greenhouse gas. Agriculture contributes significantly to its emissions. As nitrogen fertilization has been identified as one of the main sources of N_2_O, controlled application and reduction of the amount of fertilizer adapted to crop demand is essential to reduce N_2_O emissions. This requires detailed studies of the local distribution of the N_2_O emission fluxes on different croplands. Consequently, frequent spatially resolved field measurements of N_2_O concentrations are needed. A precision in the ppb range close to the ambient N_2_O level of 333 ppb is necessary. Tunable laser absorption spectroscopy using quantum-cascade lasers (QCL) as a light source is an established technique for the measurement of N_2_O traces. We present the development and validation of a compact portable setup for on-site measurement of N_2_O emissions from the soil. The setup differs from previous solutions by using an interband cascade laser (ICL), which has significantly lower power consumption compared to a QCL. The portable measurement setup allows N_2_O emission fluxes to be determined with a precision of 3.5% with a measuring duration of 10 min. The developed system enables the detection of increased N_2_O emissions because of the fertilization of fields. High N_2_O emission fluxes are indicators of the overfertilization of the field. Directly after fertilization, N_2_O fluxes between 2.9 and 5.3 µL m^−2^ min^−1^ depending on the gas acquisition site are measured during the field tests. Over time, the fluxes decrease. The obtained results compare well with data from more precise but also more complex and maintenance-intensive instruments for atmospheric research. With this system, the soil moisture as well as the air humidity and air temperature are recorded. Strong influences on N_2_O fluxes by soil moisture were observed. The presented measurement system is a contribution to the establishment of mobile N_2_O screening systems that are robust in the field and suitable for comprehensive and routine detection of N_2_O emissions from soil.

## 1. Introduction

### 1.1. N_2_O from Agriculture—A Major Greenhouse Gas

The increase in anthropogenic greenhouse gases (GHG) has an important impact on the global climate. A major source of GHG emissions is agriculture. Thus, measuring agricultural GHG emissions is necessary to understand the underlying processes in the soil and develop means to limit the cumulative GHG emissions. In a recent book (Zaman, [1]), the challenges for agricultural GHG measurements are described, and current methods for measurement of agricultural GHG emissions are reviewed. Therefore, Zaman’s book is used as the main reference in this publication. Among the three major greenhouse gases CO_2_, CH_4_ and N_2_O, the latter is mainly related to agricultural activities and the intense use of synthetic fertilizers [1]. With a global warming potential about 265 times higher than CO_2_ [2], nitrous oxide emissions make agriculture a sector with a high impact on climate change. Applying too much fertilizer leads to an excessive nitrate level in the soil. The nitrate—not needed by the plants—is converted into nitrous oxide and is emitted from the soil surface. Also, unconverted nitrate remains in the soil and leads to pollution of soil and groundwater. The aim behind a mindful usage of fertilizer is to apply only the amount of fertilizer the plants need. This protects the environment and saves resources and money while the benefit of fertilization, a high yield, remains.

Since 1990, there has been no visible trend in Europe toward a reduction in N_2_O emissions from agricultural soils [3]. To reduce fertilizer consumption, it is very important to have a clear understanding of the nitrogen cycles in the soil to know the correlation between fertilization, nutrient consumption of the plants and N_2_O emissions. To investigate these correlations, detailed studies of the local distribution of N_2_O emission fluxes on different agricultural lands as a function of mapped data such as vegetation, soil composition, moisture, etc., must first be carried out. This requires precise N_2_O measuring systems. Once the nitrogen cycles are well understood in combination with having precise measuring systems for N_2_O emissions, demand-driven fertilization can be significantly improved.

However, the determination of the N_2_O emission fluxes is a difficult task, because only a small increase in the N_2_O concentration against a natural background of 333 ppb [4] must be measured [5]. Therefore, stable calibration and high instrument precision are required. A publication by Rapson [6] describes different analytical techniques for measuring N_2_O (gas chromatography (GC), electrochemical amperometric methods and optical methods such as laser spectroscopy). GC is a low-cost technique with higher calibration requirements compared to optical methods [6]. Amperometric sensors have high sensitivity, but their disadvantage is drift, which makes long-term monitoring difficult [6,7]. Optical methods are able to obtain spectra very quickly and perform continuous measurements, making them ideal for measuring trace gas fluxes [6]. Also, the measurement of N_2_O with laser spectroscopy is very selective. At a laser emission wavelength of 4.5 µm, tests in the laboratory at negative pressure showed no cross-sensitivity to other gases in the air. Among the different approaches to measuring agricultural GHG fluxes, chamber-based methods are most frequently employed [1]. However, the use of gas collection chambers in the field may influence the emission flux compared to uncovered soil due to e.g., different temperature, humidity, etc. [1]. In a recent publication (Tallec, [8]), three different methods to measure N_2_O fluxes over an irrigated maize field were compared (automatic chamber systems, static chamber systems and two eddy current analyzers). Although this study shows that the eddy current method is a promising way to estimate and refine N_2_O budgets at the field level, it also highlights the need to continue the development of chamber methods [8]. These systems are not mobile as they cannot be moved across the field without effort. The complexity of the soil requires measurements at several sites, so a robust and mobile measurement system is a big advantage.

This paper presents the development and validation of a compact, robust, mobile, and fast laser spectroscopic system to measure N_2_O emissions fluxes on-site on the soil using chamber methods. In addition, it is easy to handle in agricultural environments.

### 1.2. State of the Art of N_2_O Measurement

For agricultural applications in the field, the mobility of N_2_O measurement systems is very important. Systems that can measure N_2_O in the concentration range relevant to agriculture are already available. One option for measuring N_2_O is using cavity ring-down spectroscopy. The Picarro G5310 is able to measure N_2_O with high precision [9]. However, this system is very expensive and with a power consumption of 200 W, it is only designed for use in static installations and not for mobile platforms. Other systems, such as GASERA’s ONE GHG, which uses photoacoustics have the accuracy necessary to resolve N_2_O concentrations in the ppb range, but their size and power consumption make them unsuitable for mobile use in the field [10]. A publication by Brümmer et al. shows the measurements of N_2_O fluxes in the field using a setup that combines the measurement system with a new automated chamber system [11]. The measuring device used for these measurements was a gas chromatograph (GC-2014, Shimadzu) on the one hand and an Aerodyne Research system using a QCL on the other. During a measurement campaign as part of the investigation by Brümmer et al. [11], flows from 5 to 270 µg N m^−2^ h^−1^ were measured using the QCL Aerodyne system. Another publication by Maier et al. presents a method to monitor the transport, production and consumption of N_2_O in soils in situ in a two-dimensional profile using a photoacoustic field gas monitor (Innova 1416, Lumasense) as a measurement system [12]. The measurement methods described by Brümmer et al. and Maier et al. are static systems. They are not suited for field measurement during cultivation as they are immobile. Recently, two systems with sample acquisition have been introduced that can be used portably in the field: the LGR-ICOS™ GLA151-N2OCM from ABB Inc. using off-axis integrated cavity output spectroscopy and the LI-7820 from LI-COR using optical feedback cavity enhanced absorption spectroscopy. The system from ABB weighs 23 kg and has a very high power consumption of 180 W [13]. With a weight of 10.5 kg and power consumption of 22 W, the LI-COR system is better suited for mobile measurements in the field [14,15]. Both optical methods are precise but quite complex and it remains to be seen whether they are robust enough for routine measurements in the field. Our system is kept as simple as possible to make it very robust. We chose a relatively short White multireflection cell [16] because this cell type has proven to be very robust and easier to adjust compared to Herriott cells [17]. A similar White cell is also part of N_2_O measurement systems for automotive exhaust gas analysis, which are commercially available from AVL [18,19].

Thus, there is still a need for mobile, robust and economically viable N_2_O measurement systems for agriculture research. This motivated us to develop the system described below.

## 2. Materials and Methods

### 2.1. Experimental Setup

The setup was developed and tested in two stages. The first stage was used to test functionality under laboratory conditions and with a permanent power supply (laboratory setup). In the second stage, the collecting, feeding and measurement of gas samples were combined into a portable system (portable setup).

In our setup, N_2_O is measured via laser absorption spectroscopy. The optical system is configurated as shown schematically in Figure 1. It consists of a distributed feedback (DFB) laser with an emission wavelength of 4.5 µm, a multi-reflection long-path cell with an optical path length of 720 cm and a volume of 300 mL and a three-stage thermoelectrically cooled MCT photodetector (Vigo System, S.A., Ożarów Mazowiecki, Poland). The laboratory setup uses a 4.52 µm DFB Quantum Cascade Laser (QCL, AdTech Optics, Los Angeles, CA, USA) as a light source. The portable setup is built with a 4.524 µm DFB Interband Cascade Laser (ICL, Nanoplus, Meiningen, Germany).

Peripherals include laser diode driver electronics (Meerstetter Engineering GmbH, Rubigen, Switzerland), which was optimized by additional in-house development. A data acquisition board (Measurement Computing, Bietigheim-Bissingen, Germany) digitalizes the detector signal. The optical stage has the dimensions 343 × 157 × 80 mm^3^ and weighs 3.75 kg. The gas flow system consists of a pump (KNF Neuberger GmbH, Freiburg, Germany) on the gas outlet side and a needle valve on the gas inlet side (Figure 2a). It was designed to regulate the absolute pressure inside the long-path cell to about 290 mbar, as well as to restrict the gas flow to approximately 1 L/min. The pressure is measured with a built-in pressure sensor (MPL3115A2, NXT Semiconductors N.V., Eindhoven, Netherlands). The stated pressure value is an empirical value coming from a trade-off between maximizing the absorption signal of N_2_O and the opposed minimizing of line-broadening effects as indicated by simulations using the HITRAN database [20].

The collection of the gas sample and the feeding into the measuring cell has been studied in two stages of development. At the first development stage, the gas sample was collected on the field and the measurement of the N_2_O concentration took place in the laboratory. The first stage was used to test functionality, as the laboratory has more constant conditions (e.g., humidity and temperature) and a permanent power supply. In the second step, the collecting, feeding and measurement of the gas sample were combined into a portable system. Therefore, the basic structure of the device was not changed. In the optical setup (Figure 1), only the QCL was replaced by an ICL and the power supply by a rechargeable battery.

#### 2.1.1. Laboratory Setup

Figure 2b shows a drawing of a bell-shaped hood that has a volume of 5 L and an area to the ground of approximately 254.5 cm^2^. The volume of the hood considered that the inside volume of the tube that sticks out of the hood is compensated by the rubber volume of the tube inside the hood. Thus, the total volume of the hood can be specified as 5 L. By setting the gas collection hood on the cropland surface, the N_2_O diffusing from the soil is captured in the hood. A built-in fan ensures the mixing of the gas inside the container. With a one-liter syringe, a part of the collected gas can be extracted and filled into sample bags (Restek, Bellefonte, PA, USA). A volume of 8.6 mL between the syringe and the valve, which consists of the ambient air is additionally transferred into the bag and results in a lower measured N_2_O concentration. This error is corrected during the data evaluation. The examination of the sample bags was done in the laboratory directly after the acquisition process. The N_2_O concentration of the sample bags is measured by plugging the outlet of the sample bags directly into the gas inlet of the measurement system, which is schematically shown in Figure 2a. Only after plugging into the gas inlet, the valve of the sample bag is opened to prevent mixing with ambient air. The underpressure generated by the pump sucks the gas sample into the measuring chamber. As soon as the gas bag is empty, it is removed from the gas inlet and the ambient air flows into the measuring chamber again. The total volume of the gas system from inlet to outlet, including measuring cell (300 mL), pump and tubes, is approx. 335 mL. This means that the sample volume is sufficient to completely flood the measuring cell and reliably determine its N_2_O concentration.

#### 2.1.2. Portable System

The portable system uses a cylindrical hood, similar to the setup used by Zaman et al. [1]. It is placed onto the crop soil surface for gas sampling (Figure 3). The hood has a volume of 8.73 L, with an area to the ground of 452 cm^2^ and a height of 19.3 cm. The N_2_O diffusing out of the soil is captured in the hood, where it leads to an increasing N_2_O concentration. The gas from the hood is pumped in a circuit, from the hood through a needle valve into the measuring cell and from there through the pump back into the hood. To the volume of the gas collection hood (8.73 L), the volume of the measuring cell (300 mL) and gas circuit (ca. 40 mL) must be added for getting the total measurement volume of 9.07 L. To record the ambient conditions, sensors for soil moisture (SMT100, Trübner GmbH, Neustadt, Germany), air humidity (SHT21, Sensirion AG, Stäfa, Switzerland) and air temperature (MPL3115A2, NXT Semiconductors N.V., Eindhoven, Netherlands) are also installed in the setup. To perform reference measurements in the field, two different N_2_O concentrations (100 ppb and 500 ppb) were filled into gas sample bags (Restek, Bellefonte, PA, USA) in the laboratory. These bags can be connected to the measurement system instead of the cylindrical chamber, similar to the setup shown in Figure 2a.

Figure 4 shows a picture of the portable setup on a cornfield. The box with a size of 20 × 43 × 33 cm^3^ and a weight of about 17 kg contains two integrated lithium-ion polymer batteries with a capacity of 4500 mAh (Hacker Motor GmbH, Ergolding, Germany). The measuring chamber is placed between the rows of corn. To ensure that the hood is tight and ambient air does not enter, additional rubber strips have been applied at the edge of the hood.

### 2.2. Measurement Method

In this work, N_2_O in air is detected via tunable diode laser absorption spectroscopy (TDLAS). Therefore, the laser emission wavelength range must be chosen according to the N_2_O photoabsorption cross section, located roughly around a wavelength of 4.5 µm which corresponds to a wavenumber of 2222 cm^−1^ (Figure 5a). The absorption cross-sections are obtained from the HITRAN database [20]. Ambient moisture (H_2_O) is also contained in the acquired samples. It has a rather strong absorption line within the spectral range of the emitted radiation. Because the H_2_O absorption line is also spectrally scanned, the ambient moisture is also measurable via its absorption. In this work, however, the humidity is determined with an external sensor (SHT12).

The laboratory setup uses a 4.52 µm DFB Quantum Cascade Laser (QCL, AdTech Optics, Los Angeles, CA, USA) as a light source. The QCL has a power consumption between 1.9 and 5.0 W depending on the laser current [21]. The maximum optical output power is given as 39.2 mW. The portable setup is built with a 4.524 µm DFB Interband Cascade Laser (ICL, Nanoplus, Meiningen, Germany). The power consumption of this laser is specified at the operating wavelength (λ_op_ = 4524 nm) as 600 mW [22]. At the operating wavelength, the optical output power is 8 mW.
Figure 5(**a**) Absorbance of N_2_O and H_2_O concentrations in synthetic air as a function of wavenumber (similar to publication [23]). (**b**) Laser current of the QCL (blue) and the acquired detector signal (green) as function of time.
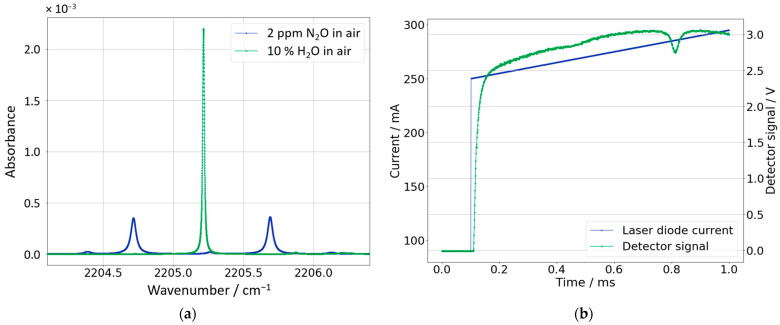



The operating temperature of the QCL was set to 38 °C. Figure 5b shows for the QCL the variation of the laser current over time that follows a ramp structure and corresponds to a wavelength change. For the ICL with the portable setup, the varying of the laser current works the same way. The custom-defined laser current was chosen to start on a constant level *I_zero_* and then to change into a ramp structure from *I_low_* to *I_high_*. To obtain a stable data acquisition with the reproducible triggering of the detector, *I_zero_* (90 mA) is intentionally set below the lasing threshold (160 mA). As for the regular data acquisition, the custom current curve was repeated with 1000 Hz. Comparing Figure 5a,b as well as regarding that the wavenumber decreases with increasing laser current, the observed absorption lines at laboratory ambient air correspond to H_2_O (at 0.8 s and/or 2205.2 cm^−1^) and N_2_O (at 0.45 s and/or 2205.7 cm^−1^).

### 2.3. Data Analysis

The data analysis in this setup is purely digital, i.e., the detector signal is fed directly into a DAC and forwarded to the PC as a tuple of “time” and “voltage” (Figure 6). To track fluctuations in the laser stability, i.e., to clearly distinguish whether the gas concentration is changing or the radiation intensity fluctuates, a reference point is chosen in the detector signal at a wavelength where no absorption occurs (Figure 6a). After acquiring a set of approx. 60 cycles, each containing an N_2_O absorption, a baseline correction is done with a 3rd-degree polynomial fit. Also included in each data cycle is an H_2_O absorption enabling the measurement of H_2_O in the air as well. However, since the air humidity was determined with an external sensor (SHT12) during the field measurements, the measurement of H_2_O via the absorption line is not mentioned further. The outcome then passes through a Butterworth (6th order, 15 kHz cut-off frequency) low-pass filter. The smoothed signal is differentiated up to the 4th order. Finally, the measured concentration of the gas species is correlated with the amplitude of the derivatives D*n*. For this purpose, the peak-to-peak value of each derivative is calculated and stored as a measurement signal (Figure 6b). The result for a measurement data point is a 6-tuple, which contains for 60 cycles the average reference value and the average amplitudes of the four derivatives. A time stamp completes the measurement data tuple. Since the average amplitude of the first derivative D1 has the lowest noise, D1 was used for the calculation of the N_2_O concentration.

## 3. Results

### 3.1. Measurement Results with Laboratory Setup

First outdoor measurements were performed on a cornfield near Schallstadt, located south of Freiburg, Germany. Sample acquisition was carried out according to Figure 2b. The first step of sampling was to place the bell-shaped container with the top valve open to avoid possible overpressure that could potentially hinder N_2_O diffusion from the soil below. After placement, the top valve was closed. This start of sampling takes 5 min. For the data evaluation of the laboratory setup, the concentration of ambient air is set to 333 ppb N_2_O as a reference before each measurement (Figure 7a). This reference value can be assumed to be approximately constant [24]. The concentration of N_2_O in ambient air has a seasonal variation of about 2 ppb. This means that an absolute measurement uncertainty of ±1 ppb due to the reference concentration leads to an assumed relative measurement uncertainty of the N_2_O concentration of almost 0.3%. Assuming a linear absorption behavior, the calculated peak-to-peak value of the first derivative D1 is then related to this reference value of 333 ppb and the zero signal in synthetic air, and an absolute concentration value is calculated.

A typical measurement curve with the laboratory setup is shown in Figure 7a. The gas samples are measured in alternation to laboratory air which serves together with the zero signal in synthetic air as a reference for the ambient N_2_O concentration (333 ppb). A slight but significant drift of around 0.2 ppb/min was corrected with a baseline fit assuming a constant N_2_O reference concentration of 333 ppb. A possible reason for baseline drift could be fluctuations in laser intensity since a normalization to a reference laser power was implemented afterward and was not included in the measurement of the reported dataset. The noise of the measurement signal in Figure 7a was determined using the ambient N_2_O reference concentration at 333 ppb. The standard deviation of the measurement signal is approx. 0.2 ppb (Table 1). Figure 7a shows the measured N_2_O concentrations from three different measurement locations on a crop field. The increased N_2_O concentration of the sample bags can be clearly seen. The result of the N_2_O concentration of a gas sample is determined via the mean value of the measured values associated with the gas sample. The error caused by noise becomes very small due to this averaging (<<0.1%) and is therefore neglected. Due to the volume of 8.6 mL of ambient air between the syringe and the valve, the dilution of the measured N_2_O concentration must also be considered. The error in volume determination is very small at 0.1 mL. The resulting relative error in the measured concentration due to dilution is less than 0.1% and therefore also neglected. The procedure from Figure 7a can be repeated every couple of days. Knowing the volume and the area to the ground of the collection hood, the N_2_O flux can be determined:(1)$$N_{2}O\;\mathrm{flux}=a_{l}\times \frac{\Delta c}{\Delta t}\;$$




$a_{l}\;:\mathrm{prefactor}\;\mathrm{laboratory}\;\mathrm{system}=0.1965\;\frac{µL}{{\mathrm{ppb}\;m}^{2}}\;$



Δc: concentration difference ppb



Δt: time difference min 




The prefactor *a_l_* is calculated from the volume of the bell-shaped hood (5 L) and its area to the ground (254.5 cm^2^). Converting to the unit µL min^−1^ m^−2^, the uncertainties related to the volume and area of the hood must be considered. The error in the volume determination is estimated to be 5 mL. The error of the area of the hood is estimated to be 2.5 cm^2^. This results in an estimated total error of 1.1% due to conversion. The total measurement uncertainty of the laboratory setup depends on the error of the ambient reference N_2_O concentration, the noise of the measurement signal, the error due to dilution and the error of the dimensions used for conversion to the unit µL min^−1^ m^−2^. The total uncertainty of the laboratory setup is estimated to approx. 1.5% using the error terms already mentioned and estimated. The determined uncertainty of the laboratory setup is only an estimate. The aim of the investigations of the laboratory setup was to test whether the measurement of N_2_O in the field is fundamentally possible.

Plotting the resulting N_2_O flux as a function of days since the last fertilizer deposition is shown in Figure 7b. The first measurements were taken two days after fertilizer deposition, where the measured N_2_O concentration is not significantly increased. After six more days, a strong concentration increase is observed. Afterward, the concentration decreases in the following measurements and converges towards the reference concentration.

### 3.2. Measurement Results with Portable Setup

Figure 8 shows the procedure of a measurement on the crop field with the portable setup at a data acquisition site. Before and after the measurement of the N_2_O diffusing from the field soil, the gas collection hood is removed from the general setup that behaves without the hood like the laboratory setup (Figure 2). Between the individual steps, the gas inlet remains open in order to get measured the N_2_O concentration of the ambient air. This allows us to detect a possible drift and correct the measured N_2_O concentration with a baseline fit. For reference, a calibration gas generator was used to fill two defined concentrations of N_2_O (100 and 500 ppb) into gas containment bags. The bags are inserted into the gas inlet tube of the system to determine the reference concentrations. Ambient air was recorded before, between and after the reference concentrations. Ambient air was also recorded before the measurement to ensure a stable system. This referencing process is also done before and after the measurement of the arable soil diffusing N_2_O. The measurement of the N_2_O diffusing out of the arable soil takes place between minute 10 to minute 20. For this, the gas collection hood is placed on the arable soil and the tubes are connected to the gas inlet and outlet (Figure 3). 

Using the signal for ambient air in Figure 8, the standard deviation of the measurement signal is determined. It is approx. 8.4 ppb. Table 1 shows the mean values and standard deviation of the measurement signal for the laboratory setup and the portable setup, each at ambient air and a different concentration.

The N_2_O concentration of the ambient air in this measurement is about 354 ppb, which is higher than the natural reference concentration of 333 ppb. This is most likely due to the measurement close to the ground during the measurement phases with ambient air. Close to the ground, the N_2_O concentration is slightly higher due to the N_2_O diffusing from the soil. The noise of the portable system is significantly higher than the noise of the laboratory setup. One reason is averaging for the laboratory setup, and another is the differences between the lasers. While the QCL of the laboratory setup has an optical output power of 39.2 mW [21], the optical output power of the ICL of the portable setup is 8 mW [22]. After the drift correction via baseline fit and the referencing with the known concentrations of 100 ppb and 500 ppb is conducted, a regression line is drawn between minutes 10 and 20. The N_2_O-flux with the unit µL m^−2^ min^−1^ is calculated from the slope of the regression line according to:(2)$$N_{2}O\;\mathrm{flux}=a_{p}\times \frac{\Delta c}{\Delta t}$$




$a_{p}\;:\mathrm{prefactor}\;\mathrm{portable}\;\mathrm{system}=0.201\;\frac{µL}{{\mathrm{ppb}\;m}^{2}}$



Δc: concentration difference ppb



Δt: time difference min 




The prefactor *a_p_* in the equation is obtained by converting ppb min^−1^ to µL m^−2^ min^−1^ using the dimensions of the hood. The prefactor *a_p_* is calculated from the total volume of the gas circuit (9.07 L) and its area to the ground (452 cm^2^). The N_2_O flux of the acquisition site shown in Figure 8 can thus be given as 1.51 µL m^−2^ min^−1^. The reference concentrations have an adjustment accuracy due to the calibration gas generator (HovaCAL^®^ digital 922-SP, Inspire Analytical Systems) that was used for mixing the gases of 2% related to the set value [25]. We assume that this accuracy error is the same for different N_2_O concentrations and, therefore, has no significant influence on the precision of the flux determination. Only the relative change of N_2_O concentration over the measuring time is relevant for the flux. Also, the determined slope of the regression line has an error of about 0.18 ppb min^−1^. This corresponds to a relative error of approximately 2.4% that influences the measured N_2_O flux. In addition, the uncertainties of the volume and area of the hood must be considered. Assuming errors in the volume and area of the hood of 10 mL and 4.5 cm^2^, the estimated total error from conversion to the unit µL m^−2^ min^−1^ is 1.1%. The total measurement uncertainty of the portable setup depends on the error of the slope of the regression line and the error of the dimensions used for conversion to the unit µL min^−1^ m^−2^. The overall uncertainty of the portable setup is estimated to be 3.5% using the error terms already mentioned and estimated. The determined uncertainty of the portable setup is only an estimate. The portable setup was developed to investigate whether measurements in the field are basically possible.

The measurement is performed for four different sites on the arable soil and is repeated every few days. This gives the result shown in Figure 9, which shows the N_2_O flux as a function of days since the last fertilizer deposition.

Figure 9 shows a large variation in the measured N_2_O emission fluxes. Therefore, it is important to measure at many points on the field to get reliable results. The N_2_O flux in Figure 9 generally decreases with increasing time intervals to the fertilization day. This makes sense because the fertilizer concentration in the soil is likely to be highest right after fertilization and, therefore, most N_2_O should be converted at this time. This means that the N_2_O flux out of the soil is highest right after fertilization and decreases over time. Day 26 is an exception. Here, the flux is very high compared to the days before and after. A possible reason is the soil moisture that is measured with the external sensor in parallel to the N_2_O flux (Figure 10). From day 26, the soil moisture increases. This also fits very well with the weather at the time when these measurements were taken. Initially, it was very hot and dry and between days 14 and 21, it rained, which explains the increased soil moisture. Increased soil moisture promotes the dissolution of the fertilizer and its distribution into the soil. This means that the fertilizer can be converted faster and at the same time more N_2_O is produced. Acquisition site 4 does not show this high flux on day 26, although all sites were fertilized equally. This also demonstrates the importance to measure at many sites in order to make statistically reliable statements.

## 4. Discussion

The N_2_O measuring system was implemented and characterized, first as a laboratory setup and then as a mobile and portable setup. It was shown that the N_2_O concentration of the collected gas from the croplands could be measured and the N_2_O flux diffusing out of the soil determined. The test measurements with the laboratory setup show a high N_2_O flux a few days after fertilization. With increasing time, the N_2_O flux decreases again. This correlation is generally also observed with the portable setup. Dry soil in the first two weeks and wet soil afterward explain the higher flux after these first two weeks. In the first measurements after fertilization, the N_2_O flux is between 2.9 and 5.3 µL m^−2^ min^−1^ depending on the gas acquisition site. To make a comparison with other investigations [1,11] easier, the flux is given in nitrogen equivalents (µg N m^−2^ h^−1^). This can be calculated using the ideal gas equation. Assuming an air pressure of 1 bar and with the measured temperature around 30 °C, the N_2_O flux is between 193 and 354 µg N m^−2^ h^−1^ depending on the gas acquisition site. This roughly corresponds to the values in Zaman et al. [1] where the N_2_O emissions during a 35-day measuring campaign without fertilization are between 5 and 245 µg N m^−2^ h^−1^. The values determined with the Aerodyne Research system using a QCL laser [11] are also in this range (5 to 270 µg N m^−2^ h^−1^).

The noise of the measurement signal is quantified by its standard deviation. The standard deviation is about 0.2 ppb for the laboratory setup and about 8 ppb for the portable setup. The reasons are different averaging and optical output powers of the lasers. While the QCL of the laboratory setup has an optical output power of 39.2 mW [21], the optical output power of the ICL of the portable setup is 8 mW [22]. The uncertainty of the laboratory setup using a QCL can be estimated at 1.5%, this applies to N_2_O concentration and N_2_O flux. The uncertainty of the portable setup using an ICL for the N_2_O flux out of the arable soil is assumed to be 3.5%. The Aerodyne Research system, which includes a QCL, has an uncertainty of 0.2% [11]. The uncertainty of our system is thus higher by approx. the factor 17.5. The Aerodyne Research system has a 76 m long path cell which is more than 10 times longer than in our setup (7.2 m). Also, the volume of the Aerodyne measurement cell is 500 mL, which is higher than for our setup (300 mL). These are reasons for the lower uncertainty of the Aerodyne Research system. For our setup, we chose a relatively short White multireflection cell [16], because this cell type has proven to be very robust and easier to adjust compared to Herriott cells [17] as incorporated in the Aerodyne Research system. A similar White cell is also part of N_2_O measurement systems for automotive exhaust gas analysis, which are commercially available from AVL [18,19].

The setup presented here is compact, robust and mobile. In addition, an ICL is installed in the portable setup, resulting in low power consumption. All in all, this makes the developed system mobile and easier to use in the field compared to state-of-the-art solutions.

## 5. Conclusions

To carry out complex screening measurements on site, the measuring system must be portable and robust. With the presented mobile system, it is possible to record the N_2_O emissions step-by-step on a whole field. The weight and the size of the setup are kept low (20 × 43 × 33 cm^3^, 17 kg), and it is powered by a battery pack. To guarantee the longest possible measuring time, the power consumption should be as low as possible. For this reason, a more energy-saving ICL is used in the portable setup instead of a QCL. The gas sampling unit is integrated into the measuring system. The N_2_O concentration in the measuring chamber accumulates with increasing time. Since the gas is pumped in a circuit, the gas constantly flows through the collection hood and the long-path cell. This ensures that only gas enters the measuring chamber by natural diffusion and not by suction. This is important to keep the measurements as close as possible to the natural conditions. Environmental parameters, especially soil moisture, strongly influence the N_2_O flux. The recording of these parameters—not only the soil moisture but also the air humidity and air temperature—is therefore very important.

As a further development step, the measuring device could be integrated into a field robot application with other sensors and measurement systems. In addition to the N_2_O concentration, the NH_3_ emission flux is also an important parameter. This parameter could also be integrated into a future system. To get a better understanding of the correlation between the emission of N_2_O and the amount of fertilizer on the field, as well as the chemical and physical processes in the soil, still, many more measurements have to be performed on different fields and under different conditions.

## Figures and Tables

**Figure 1 sensors-23-06686-f001:**
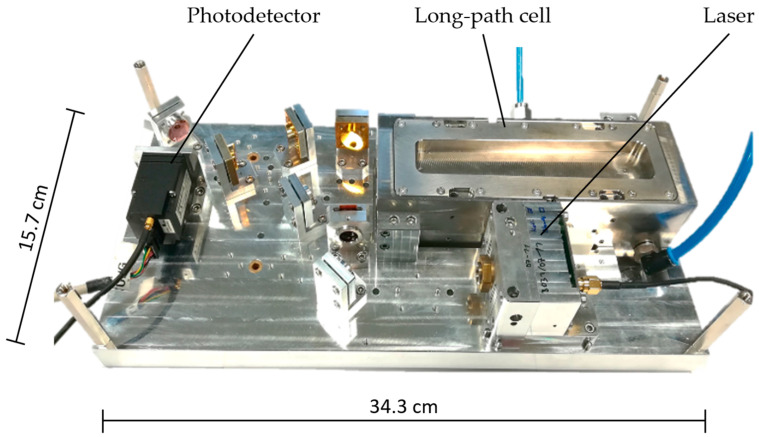
Optical measurement setup including laser with an emission wavelength at 4.5 µm, 720 cm long-path measurement cell and thermoelectrically cooled MCT photodetector. This optical setup was used in both development stages with a QCL for laboratory setup and an ICL for portable setup. N_2_O is measured via absorption spectroscopy.

**Figure 2 sensors-23-06686-f002:**
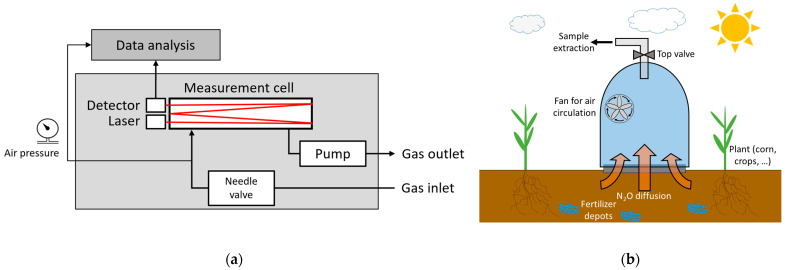
(**a**) Schematic view of the laboratory measurement setup. (**b**) Drawing of the bell-shaped hood for gas sampling and extraction. Gas samples were taken from the hood for laboratory measurements.

**Figure 3 sensors-23-06686-f003:**
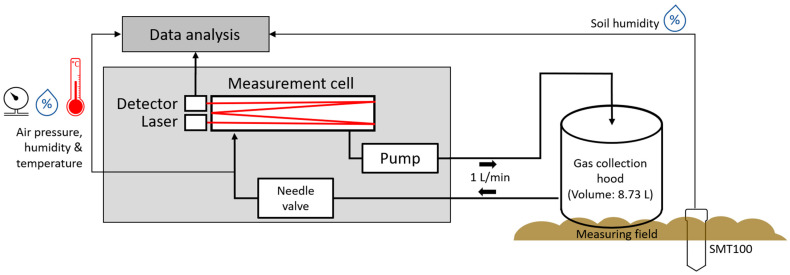
Schematic view of the portable measurement setup. Gas samples are measured directly on the field.

**Figure 4 sensors-23-06686-f004:**
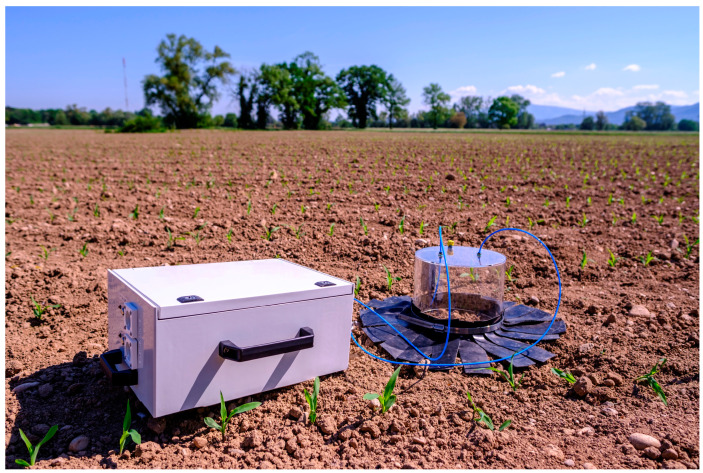
The portable system on a cornfield.

**Figure 6 sensors-23-06686-f006:**
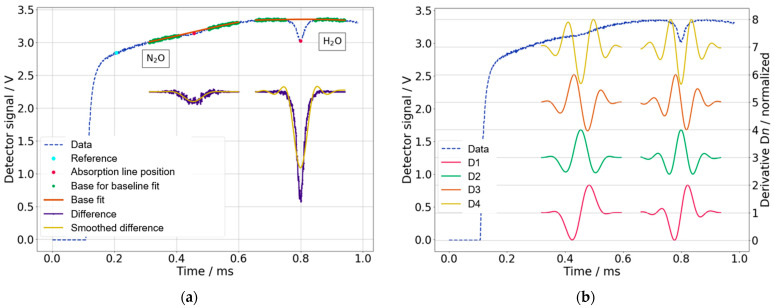
(**a**) Data processing steps: raw data acquisition, absorption line position, baseline fit and subtraction, Butterworth filter smoothing. (**b**) Normalized derivatives D1 to D4. The left/right derivatives accordingly correspond to N_2_O/H_2_O.

**Figure 7 sensors-23-06686-f007:**
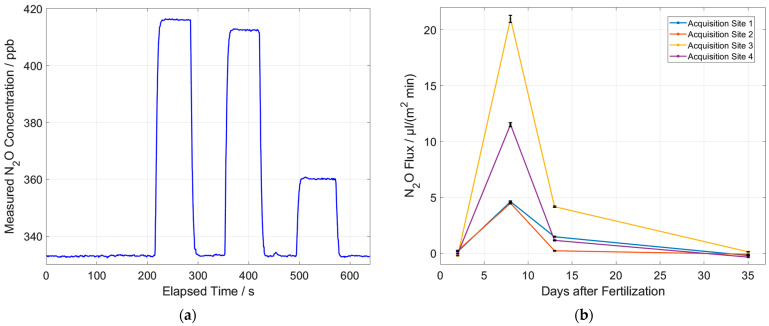
(**a**) Laboratory measurement of gas samples from three different measurement locations on a crop field. (**b**) Calculated flux for four different positions on a crop field as function of days since the last fertilization occurred. The measurement errors shown in the figure are an estimate.

**Figure 8 sensors-23-06686-f008:**
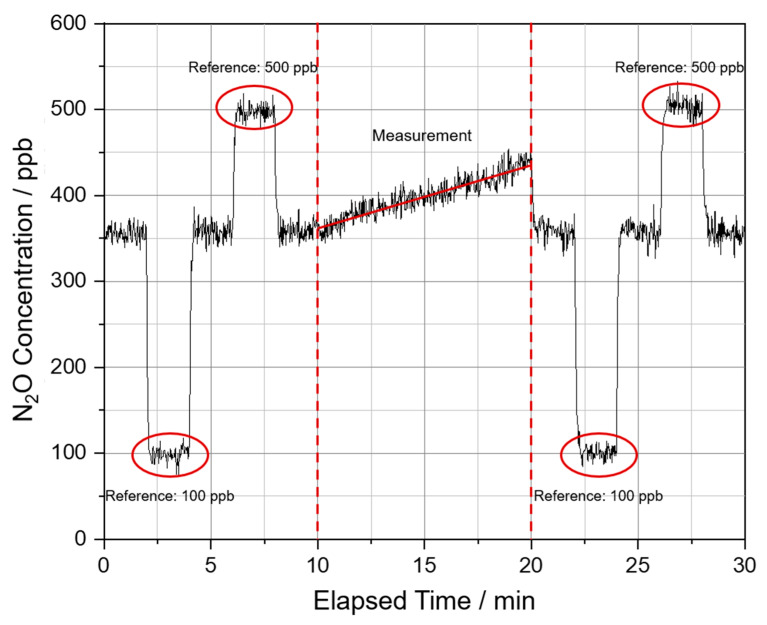
Measurement at a data acquisition site on the cornfield.

**Figure 9 sensors-23-06686-f009:**
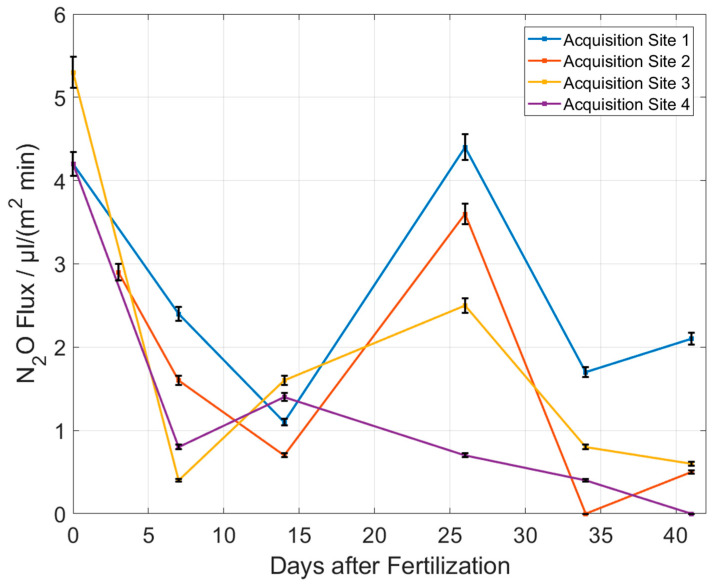
Measured N_2_O emission fluxes for different positions on a crop field as function of days since the last fertilization occurred. The measurement errors shown in the figure are an estimate.

**Figure 10 sensors-23-06686-f010:**
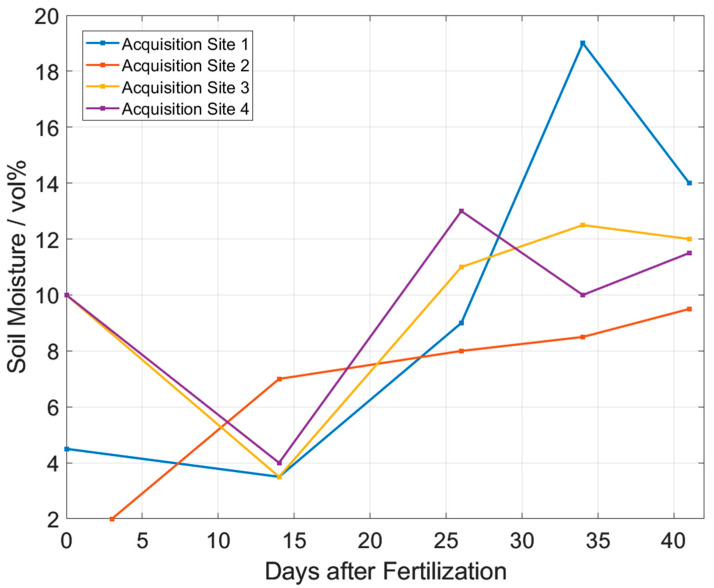
Measured soil moisture corresponding to measured N_2_O emission fluxes.

**Table 1 sensors-23-06686-t001:** Comparison of mean values and standard deviation of the measurement signal for the laboratory setup and the portable setup, each at ambient air and a different concentration.

	Lab. Setup (amb. Air)	Lab. Setup (1st Sample)	Port. Setup (amb. Air)	Port. Setup (Ref.: 500 ppb)
Mean Value	333.0 ppb	416.1 ppb	354.3 ppb	497. 5 ppb
Standard Deviation	0.2 ppb	0.2 ppb	8.4 ppb	8.2 ppb

## Data Availability

The underlying measurement data are not publicly available and can be requested from the authors if required.

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
