# Peer review of "A Portable Laser Spectroscopic System for Measuring Nitrous Oxide Emissions on Fertilized Cropland"

_sensors, 2023, doi:10.3390/s23156686_

Round 1
Reviewer 1 Report
The paper reports the design and employment of a (or maybe two) spectroscopic systems to measure N2O emitted from soil, fertilized croplands. The spectrometer is designed to be operational on the field site, equipped with an ICL. Another instrument (of the same design) is employed in the lab but operates a QCL.
The paper is well written and clearly structured. However, to me, it is stays unclear, why the authors have developed two instruments and to which instant both or only the field-deployable shall be in the focus of the paper. From the lab setup I cannot infer a single added value, neither for the reader nor for the application. The level of experimental details the authors provide is really valuable – the level of details how the authors have analysed the measured data not that much.
My recommendation is ask the authors to resubmit the paper with major revisions employed according to the issues listed in the file.

Author Response
Cover letter for revised manuscript Sensors-2470506:
A portable laser spectroscopic system for measuring nitrous oxide emissions on fertilized cropland
Dear editors,
We are submitting a revised version of our manuscript Sensors-2470506.
We have responded to all comments and recommendations of the reviewers. The suggestions of the reviewers were very helpful, and we are convinced that the manuscript has improved a lot by following the recommendations.
We would greatly appreciate if the revised manuscript now can be accepted for further processing and publication in Sensors.
Yours sincerely,
Gerrit Stiefvater
First, we like to thank the reviewer for the positive general evaluation of our manuscript and the careful review. Following your recommendations, we hope we could substantially improve the quality of the manuscript.
In the following we respond to your suggestions in detail:
Because it was unclear, why in the paper are described two setups: The setup was developed and tested in two stages. The first stage was used to test functionality, as the laboratory has more constant conditions (e.g. humidity and temperature) and a permanent power supply. In the second step the collecting, feeding and measurement of the gas sample was combined to a portable system. Therefore, the basic structure of the device was not changed. In the optical setup, only the QCL was replaced by an ICL and the power supply by a rechargeable battery. In the revised script, we tried to make this more understandable.
- p. 8, Fig. 7 (a): Please quantify the noise before and behind the sample peak at 500-600 s and explain why is not there before and would it be there afterwards – basically it doubts whether the system is stable on a 10-min scale.
The noise is also there before the sample peak at 500 – 600 s. The reason for the lower N2O concentration is a drift. In the revised manuscript, the data were re-evaluated. The new Figure 7 (a) already shows the data where the drift has already been corrected. Using the new plot, the noise was determined via the ambient reference at 333 ppb. Likewise, Table 1 below Figure 8 compares the noise of the laboratory setup with the noise of the portable setup (line 306 ff and Table 1).
- p. 8, lines 252/253: Please, most reliable data should be worth to be shown here instead! Why showing a premature data set if a better one would be available?
The graph has been adjusted. The new Figure 7 (a) already shows the data where the drift has already been corrected.
- p.8, line 259-261: Seems that a real calibration process has not been done - so measurement uncertainties were not estimated state-of-the-art. Neither the assumed reference concentration of N2O in air has been inferred and uncertainties associated, nor the actual spectroscopic measurement been analysed state-of-the-art. So, authors need to explain here reasons for doing so or improve.
Together with the ambient N2O concentration the zero signal was used as a reference. This is not a true calibration. The data were only related to the reference concentration of 333 ppb and the zero signal. A rough error estimate was re-performed considering additional error terms and is available in the revised manuscript. The aim of the investigations of the laboratory setup was only to test whether the measurement of N2O in the field is possible in principle. (line 289 ff and line 298 ff)
- p.8, line 265: Please spend the equation how the flux has been inferred and which real measurement data have been used to execute the calculation.
The same equation was used also below for the portable setup, but with a different prefactor. The equation was adjusted and added to the manuscript. (line 319 ff)
- p.8, line 265-268: What's the accuracy of the inferred fluxes now? Before the concentration measurement uncertainty has not been properly estimated, and now some additional influence quantities are entering the equation to calculate the flux which also seem to have no uncertainties with them - sorry, it might all okay, but without showing any data/estimation, I cannot know whether flux differences in Fig. 7 (b) might be simply due to some random scattering? Even then, what Fig. 7 (b) shows is only a single data point with an apparent change of the inferred fluxes – what conclusion can be drawn from a single point data set?
The aim of the investigations of the laboratory setup was only to test whether the measurement of N2O in the field is possible in principle. Only a rough error analysis was carried out. This was revised in the manuscript. Among other things, the error term that arises when converting to the other unit was considered. To show that the flux values vary, i.e. that it is not a random scatter, error bars have also been inserted in Fig. 7 (b). This variation shows the importance to measure at many points on the field to get reliable statistics. (line 326 ff)
- p.9, line 285-288: This actually proves that all the assumptions with the stable N2O in air concentration and its estimated uncertainty of 1 ppb is not well justified experimentally.
The concentration in the laboratory and on the field is not comparable. The measurement on the field is close to the ground during the measurement phases with ambient air. The measurement in the laboratory is carried out under room conditions. Close to the ground, the N2O concentration is slightly higher due to the N2O diffusing from the soil. This is described in the revised manuscript under table 1 (line 377 ff)
- p.9, line 293: Regression analysis- What are the parameters of this regression analysis or has just equation (1) been applied?
Regression analysis was done for the revision with the data between minute 10 and 20. The error of the regression analysis is given in the revised manuscript. (line384 ff)
- p. 9, equation (1): What about the uncertainty of this factor plus the uncertainty?
Rough error analysis was carried out for the revision and the uncertainty of the prefactor is now considered. (line 407 ff)
- p. 9, line 301/302: That might be true, but does not prevent a proper analysis ... so, e.g., if biases were different for different concentration levels, in turn it would result in different slopes of the regression line.
We assume that the biases are same for different N2O concentrations and the accuracy error therefore has no significant influence on the precision of the flux determination. Only the relative change of N2O difference concentration over the measuring time is relevant for the flux. The paragraph has been revised again to make the assumption understandable (line 399 ff).
- p.9, line 302: “… manual fitting of the regression line …” - Not sure what's behind a "manual fitting of a regression line"? Isn't it either a regression line (calculated) or manually drawing (a line) eye-led?
Regression analysis was done (calculated) for the revision with the data between minute 10 and 20. The paragraph has been revised to remove ambiguity regarding the regression line and uncertainties (line 387 ff and line 403 ff).
- p. 9, line 303/304: “… regression is subject to errors.” – So, please, quantify them.
The error of the slope of the regression line is quantified and given in the next sentence. It was determined more precisely again and added to the revised manuscript (line 403 ff.).
- p. 9, line 304: “maximum error of about 1 ppb/min” Where does this value really come from?
The measurement uncertainty was determined more precisely again (line 403 ff.). It is the error of the slope of the regression line. However for a reliable error analysis, significantly more field measurements need to be carried out.
- p. 9, line 306: The statement lacks supporting experimental (or reasonable estimated) evidence and need to be justified by data.
This is only a rough estimate of the error. The estimation of the error was revised in the manuscript. The portable setup was developed to investigate whether a measurement in the field is basically possible. That’s why only a rough error analysis was carried out.
- p. 11, 341-344: Comparing what here? Apples can be green to red, bananas can be green to yellow, so some apples must have been bananas in between? Don't understand what these coarse similarities ill-defined numbers should prove?
The portable setup was developed to investigate whether a measurement in the field is basically possible. A comparison with another system should show that our measuring system is in approximately the same range. This is to show that a measurement with this system is possible in principle under the conditions in the field.
Some additional minor issues:
- p.2, line 65: cavity ring-down spectroscopy rather than spectrometer
The manuscript has been modified accordingly
- p.3, Fig. 1: Is this the field-deployable setup with the ICL or the laboratory one with the QCL - better to state in the Fig. caption
The figure shows the general structure. As this is a general description of the measurement setup, no laser designation has been included.
- p.8, Fig. 7: caption says (b) displays concentrations, axes title says fluxes instead.
The manuscript has been modified accordingly.
- p. 8, line 279: Where does the defined value come from and how about changes due to sample bag influence when transferring gas in and out?
The manuscript has been modified accordingly and a more detailed description added.
- p. 9, equation (1): the formulation of this equation is not very scientific and is not according to standards how to express quantities and equations.
The manuscript has been modified accordingly.
- p. 11, line 340: I understand here micro-gram Newton per square meter and hour. Why shall one use this unit for fluxes?
N is for Nitrogen. The unit describes how many micrograms of nitrogen atoms diffuse out of the soil per square meter per hour. This is a unit used also in other publications and is used for comparison. The unit is explained more clearly in the revised manuscript.
Additional information: According to the second reviewer, the keywords (line 33) were arranged alphabetically and the chapter “State of the art” was placed in the introduction.

Reviewer 2 Report
This manuscript (sensors-2470506) presents a compact, portable setup using Interband Cascade Laser for precise on-site measurement of Nitrous Oxide (N2O) emissions from soil, an important greenhouse gas heavily emitted by agriculture. This system also captures soil moisture, air temperature, and humidity data, contributing to the development of robust, mobile N2O screening systems for regular soil emission detection. The manuscript is well-written, detailed, very interesting, and suitable for acceptance. The figures and captions are appropriate. I only suggest changes according to the following comments.
Keywords in alphabetic order;
The introduction needs improvement. Talk about measurement methods. Why is the laser better than other methods? Does N2O measurement overlap with other gases? What are the quantification problems faced? Application for environments needs improvement. For example, topic 2.1 of the material and methods should be associated with the introduction, not the materials and methods session.
Add references to support your introduction and discussion.
Best regards
Please check the sentences for minor grammatical errors.
Author Response
Cover letter for revised manuscript Sensors-2470506:
A portable laser spectroscopic system for measuring nitrous oxide emissions on fertilized cropland
Dear editors,
We are submitting a revised version of our manuscript Sensors-2470506.
We have responded to all comments and recommendations of the reviewers. The suggestions of the reviewers were very helpful, and we are convinced that the manuscript has improved a lot by following the recommendations.
We would greatly appreciate if the revised manuscript now can be accepted for further processing and publication in Sensors.
Yours sincerely,
Gerrit Stiefvater
Response to Reviewer Report #2
First, we like to thank the reviewer for the positive general evaluation of our manuscript and the careful review. Following your recommendations, we hope we could substantially improve the quality of the manuscript.
In the following we respond to your suggestions in detail:
1. Keywords in alphabetic order:
Keywords have been sorted accordingly
2. The introduction needs improvement. Talk about measurement methods. Why is the laser better than other methods? Does N2O measurement overlap with other gases? What are the quantification problems faced?
In the introduction, more information has been added about different measurement methods and it has been briefly explained why laser spectroscopy offers the best possibilities for the application shown here.
3. Application for environments needs improvement. For example, topic 2.1 of the material and methods should be associated with the introduction, not the materials and methods session.
Chapter “State of the art” are now associated with the introduction.
4. Add references to support your introduction and discussion.
Some references were added.
5. Comments on the Quality of English Language: Please check the sentences for minor grammatical errors.
Sentences were checked, some minor grammatical errors were found and improved.
Additional information: According to the second reviewer, some changes, especially concerning the estimation of measurement uncertainties, were made.

Round 2
Reviewer 1 Report
The paper has improved according to my understanding of the subject. The authors have responded to my concerns very well. Happy to see this paper published. However, my last comment still would be that SI units should have been used, at least explained once. Knowing that many papers don't do that, I accept the common sense of the community.